

# A systematic comparison of ACE-FTS $\delta$D retrievals with airborne in situ sampling

Benjamin W. Clouser[1], Carly C. KleinStern[2], Adrien Desmoulin[1], Clare E. Singer[1], Jason M. St. Clair[3,4], Thomas F. Hanisco[3], David S. Sayres[5], and Elisabeth J. Moyer[1]

[1]Department of the Geophysical Sciences, University of Chicago, Chicago, IL, USA
[2]Department of Physics, University of Chicago, Chicago, IL, USA
[3]Atmospheric Chemistry and Dynamics Laboratory, NASA Goddard Space Flight Center, Greenbelt, MD, USA
[4]GESTAR-II, University of Maryland Baltimore County, Baltimore, MD, USA
[5]Harvard John A. Paulson School of Engineering and Applied Sciences, Harvard University, Cambridge, MA, USA

**Correspondence:** Benjamin W. Clouser (bclouser@uchicago.edu)

**Abstract.** The isotopic composition of water vapor in the upper troposphere and lower stratosphere (UTLS) can be used to understand and constrain the budget and pathways of water transport into that region of the atmosphere. Measurements of the water isotopic composition help further understanding of the region's chemistry, radiative budget, and the sublimation and growth of polar stratospheric clouds and high-altitude cirrus, both of which are also important to stratospheric chemistry and
Earth's radiation budget. Here we present the first intercomparison of water isotopic composition ($\delta$D) using in situ measurements from the ChiWIS, Harvard ICOS, and Hoxotope instruments and satellite retrievals from ACE-FTS. The in situ data comes from the AVE-WIIF, TC4, CR-AVE, StratoClim, and ACCLIP field campaigns, and satellite retrievals of isotopic composition are derived from the ACE-FTS v5.2 data set. We find that in all campaign intervals, the satellite retrievals above about 14 km altitude are depleted by up to 150 ‰ with respect to in situ measurements. We also use in situ measurements from the
ChiWIS instrument, which has flown in both the Asian Summer Monsoon (AM) and the North American Monsoon (NAM), to confirm the isotopic enhancement in $\delta$D observed in satellite retrievals above the NAM.

## 1  Introduction

The abundance of water vapor in the stratosphere is a critical control on ozone production and destruction, surface climate, and stratospheric temperatures (Shindell, 2001). Methane and water oxidation are the primary sources of the hydroxyl radical,
which helps to control ozone in the lower stratosphere (Stenke and Grewe, 2005). Moisture concentrations also provide strong controls on the distribution and frequency of polar stratospheric clouds (PSCs) and high-altitude cirrus, both of which provide surfaces on which heterogeneous chemical reactions occur (Zondlo et al., 2000). Furthermore, of the molecules responsible for the greenhouse effect, water vapor makes the largest direct contribution (Held and Soden, 2000), and stratospheric water plays a disproportionate role (Shindell, 2001; Dessler et al., 2013). The transport of water into and through the upper troposphere
and lower stratosphere (UTLS) is of critical importance to our understanding of current and future climate, and the isotopic composition of that water can provide a needed constraint on this transport process.





Observations of water isotopologues in Earth's atmosphere provide unique information about an air parcel's condensation, sublimation, and mixing history (Webster and Heymsfield, 2003; Galewsky et al., 2016). As hydrometeors grow in an ascending and cooling air parcel, they take up water vapor. Since the heavier isotopologues of water (e.g., HDO and $H_2^{18}O$) have lower

vapor pressures than $H_2^{16}O$, they are preferentially taken up during growth, thereby leaving the vapor isotopically depleted and the condensate isotopically enriched. The isotopic composition of water vapor can therefore provide an important observational constraint to identify different sources of water vapor to the UTLS and to improve our understanding of the microphysical processes impacting cirrus cloud formation. In the last decade, in situ measurements obtained from airborne platforms have become sufficiently accurate and precise to allow interpretation of their substantial temporal and spatial variations. However,

isotopic signatures in water vapor are not straightforward to interpret, as they are the result of both complex microphysical processes and larger-scale dynamical processes. These observations give unprecedented detail into the importance of convective influence on stratospheric water vapor in the mid-latitudes.

Deep convection is an important transport pathway for aerosols, trace gases, and pollutants from Earth's boundary layer into the stratosphere. The key sources of water vapor to the stratosphere are predominately due to large-scale ascent and dehydration

of air as it passes through the tropical tropopause layer (TTL), and in-situ production from methane oxidation. In addition, convective events and volcanic eruptions can directly inject water vapor into the stratosphere from the troposphere. Both remote sensing (Nassar et al., 2007; Moyer et al., 1996) and in situ (Hanisco et al., 2007; Sayres et al., 2010) instruments have measured isotopic profiles that show isotopic enrichment with increasing altitude, indicating the importance of convectively lofted ice to the UTLS water budget. Khaykin et al. (2022a) observed strong isotopic enhancement of $H_2O$ and HDO in the stratosphere

due to the Hunga Tonga eruption. The Asian Summer Monsoon (AM) and North American Monsoon (NAM), which are annual changes in circulation patterns characterized by significant convection, are climatically significant contributors to stratospheric water vapor and have understandably been the focus of two recent NASA campaigns: ACCLIP and DCOTSS, respectively. The AM may contribute up to 75% of the upward water vapor flux to the tropopause in Northern Hemisphere summer (e.g. Gettelman et al. (2004); Kremser et al. (2009)), making it a particularly important region for UTLS isotopic studies. Analysis

of ACE-FTS satellite data (Randel et al., 2012) shows significant differences in water vapor isotopic enhancement between the North American and Asian monsoons, suggesting differences in water transport processes, but until now, no in-situ water isotopologue measurements in the AM have tested this observation.

Despite the key role stratospheric water vapor plays in both the radiation budget of the Earth and the chemistry of the stratosphere, current climate models struggle to predict water vapor in the lower stratosphere. There is a significant bias across

the ensemble of models in CMIP6, with models showing a substantial moist bias compared with observations (Keeble et al., 2021; Charlesworth et al., 2023). Almost all climate models predict that stratospheric water vapor is likely to increase with increased $CO_2$. A better understanding of current and future changes in stratospheric water vapor concentrations requires stronger observational constraints on the importance and variability of different sources of water vapor to the stratosphere. Furthermore, isotopically-enabled GCMs do a poor job simulating both water content and isotopic composition of water in the

UTLS region (Eichinger et al., 2015).



Several satellites have observed vapor phase HDO and $H_2O$ in the UTLS of Earth's atmosphere in recent decades. The Atmospheric Trace Molecule Spectroscopy (ATMOS) Fourier-transform spectrometer observed HDO and $H_2O$ in Earth's atmosphere between 100 mb and 10 mb ($\approx$15 to 30 km) (Farmer, 1987; Irion et al., 1996). These observations were inter-mittantly made from the Space Shuttle via solar occultation during four missions between the years of 1985 and 1994. The sub-millimeter radiometer (SMR) aboard the ODIN satellite measured $H_2O$, $H_2^{18}O$, $H_2^{17}O$, and HDO in Earth's Stratosphere and Mesosphere from 2001 through the present day (Murtagh et al., 2002; Zelinger et al., 2006; Murtagh et al., 2020). The Envisat satellite (Louet and Bruzzi, 1999) contained a Fourier transform spectrometer for the detection of limb emission spectra in the middle and upper atmosphere called the Michelson Interferometer for Passive Atmospheric Sounding (MIPAS). This instrument observed HDO and $H_2O$ profiles at altitudes above about 10 km from July 1, 2002 through April 8, 2012 (Fischer et al., 2008; Steinwagner et al., 2007). The Atmospheric Chemistry Experiment Fourier Transform Spectrometer (ACE-FTS) (Bernath et al., 2005)observes $H_2O$, $H_2^{18}O$, $H_2^{17}O$, and HDO via solar occultation. The instrument has been in operation from 2005 through the present day, and measures water vapor and its isotopologues from the lower troposphere up to approximately 50 km, although the measurement is highly sensitive to the presence of thick clouds (Boone et al., 2005).

There are few studies in the literature comparing retrievals of water vapor isotopic composition from different instruments in field conditions. Lossow et al. (2011) compared HDO retrievals from the Envisat/MIPAS instrument to those of the ODIN/SMR and ACE-FTS instruments. The ACE-FTS data in this study came from v2.2 retrievals covering January to March of 2004. This work found large disagreements below about 15 km, although latitudinal structures in HDO amount were consistent. There was some agreement in the 15-20 km range, and fairly good agreement above 20 km. In general, MIPAS and ACE-FTS agreed to within 10%, and MIPAS showed higher HDO abundances than ACE-FTS. Both instruments show considerably more than Odin/SMR. Observed biases were consistent with uncertainties in spectroscopic parameters.

More recently, De Los Ríos et al. (2024) compared $H_2O$ and HDO from two retrievals from Envisat/MIPAS satellite with ACE-FTS over the common interval from February 2004 to April 2012. They compare the MIPAS-IMK V5, MIPAS-ESA V8, and ACE-FTS v4.1/4.2 retrievals using a profile-to-profile approach as well as by comparing climatological structures. Stratospheric profiles of $H_2O$ retrievals show good agreement between 16 and 30 km, with biases between profile-to-profile comparisons near zero for the MIPAS-IMK and ACE-FTS data sets. However, the HDO and $\delta D$ retrievals from MIPAS-ESA and ACE-FTS exhibit low biases compared to MIPAS-IMK (typically -41.2 ‰ to 10.5 ‰).

The work of St. Clair et al. (2008) included a comparison of the isotopic compositions observed by the Hoxotope and Harvard ICOS instruments which both flew aboard NASA's WB-57F research aircraft during the AVE-WIIF campaigns (see section 4.1 for more details on AVE-WIIF). This intercomparison showed agreement in $H_2O$ measurements over three orders of magnitude between Hoxotope, Harvard ICOS, and the Harvard Water Vapor (HWV) Lyman-$\alpha$ instrument (Weinstock et al., 1994). A line fit to the Hoxotope and HWV $H_2O$ retrievals yields a slope of 1.00, an intercept of 0.96 ppmv, and an $R^2$ value of 0.98. The HDO values retrieved by Hoxotope and Harvard ICOS during the AVE-WIIF campaign agree to within their stated accuracies over the full range. A line fit to the HDO retrievals yields a slope of 1.05, and intercept of -0.14 ppbv, and an $R^2$ value of 0.99. The Harvard ICOS instrument measures higher than Hoxotope on average, but is still within the combined uncertainty of the instruments.





Hanisco et al. (2007) contains a brief intercomparison of the $\delta$D values retrieved by Hoxotope and Harvard ICOS in the AVE-WIIF campaign, stating that the average absolute difference between the instruments was 15 ‰, well within the stated uncertainties of 50 ‰.

In situ and satellite data sets of UTLS isotopic composition can provide valuable constraints on GCMs, and further our understanding of water transport into the UTLS. As a first step towards imposing more global constraints, we present here the first intercomparison between in situ and satellite measurements using the Harvard ICOS, HOxotope, and Chicago Water Isotope Spectrometer (ChiWIS) in situ data sets and satellite retrievals from the Atmospheric Chemistry Experiment Fourier Transform Spectrometer (ACE-FTS).

## 2 Definitions

In this work we compare the in situ measurements of ChiWIS, Harvard ICOS, and Hoxotope to the ACE-FTS retrievals. We compare three quantities provided by each instrument: $H_2O$, HDO, and $\delta$D. The isotopic composition, $\delta$D, is the fractional deviation in per mil (‰) units of the observed D/H ratio from that of a known standard:

$$\delta D = \left( \frac{R}{R_{\mathrm{SMOW}}} - 1 \right) \times 1000, \tag{1}$$

where $R_{\mathrm{SMOW}} = 155.76 \times 10^{-6}$ is the isotopic ratio of [D] to [H] in Vienna Standard Mean Ocean Water (SMOW, HAGE-MANN et al. (1970)). To write the isotopic ratio in terms of measured quantities, we use the approximation:

$$R = \frac{[\mathrm{D}]}{[\mathrm{H}]} = \frac{[\mathrm{HDO}] + 2\,[\mathrm{D_2O}]}{2\,[\mathrm{H_2O}] + [\mathrm{HDO}]} \approx \frac{[\mathrm{HDO}]}{2\,[\mathrm{H_2O}]}. \tag{2}$$

$\delta$D notation is often used because it is insensitive to how the isotopic ratio is defined: [D]/[H] or [HDO]/[H_2O] both yield the same $\delta$D values. This allows for easier and more universal comparison across different definitions.

An airmass is said to be isotopically depleted with respect to another if its isotopic composition is more negative, and isotopically enhanced if its isotopic composition is more positive. For reference, typical $\delta$D values in the UTLS are about -500‰, with significant regional and seasonal variation.

## 3 Instrument Descriptions

In this study we compare the in situ measurements of two airborne off-axis integrated cavity output spectrometer (OA-ICOS) instruments, the Chicago Water Isotope Spectrometer (ChiWIS) (Clouser et al., 2024) and Harvard ICOS (Sayres et al., 2009), and an in situ laser-induced fluorescence (LIF) instrument, Hoxotope (St. Clair et al., 2008), with satellite retrievals from the Atmospheric Chemistry Experiment Fourier Transform Spectrometer (ACE-FTS) (Bernath et al., 2005). Figure 1 summarizes the spatial and temporal extent of the data considered here. Flight tracks from the ChiWIS instrument (cyan) and Harvard ICOS and Hoxotope instruments (lime green) are shown for each campaign, as well as the averaging regions (black boxes) used in subsequent sections for the ACE-FTS instrument. Altogether, these measurements cover the latitude range of 10° N to 60° N,



**Figure 1.** ACE-FTS retrievals of $\delta$D at 16.5 km altitude during the (a) boreal summer months (JJA) and (b) boreal winter months (DJF). Superimposed on both contour plots are the flights tracks of the ChiWIS instrument (cyan) and the Harvard ICOS/Hoxotope instruments (lime green). The black boxes show the spatial boundaries within which ACE-FTS occultations are collected for comparison to the relevant field campaign. Latitude-longitude range of each box is listed in Table 1.

120  including the NAM and AM systems, the tropics, subtropics, and mid-latitudes from about 12 km to 20 km in altitude. Figure 1 shows ACE-FTS climatologies for the years 2004-2022 from the boreal summer (JJA) and boreal winter (DJF) seasons. The isotopic enhancement (see section 2 for definition) over the NAM relative to the AM is highly apparent in the boreal summer averages, as are the extreme depletions found in the tropics during boreal winter.





### 3.1 ChiWIS

ChiWIS is an OA-ICOS instrument which has to date flown in the StratoClim (2017) and ACCLIP (2021/2022) field campaigns. The instrument uses a tunable diode laser (TDL) to rapidly scan over $H_2O$ and HDO absorption features centered around 2647.6 nm. The highly reflective mirrors of the instrument's optical cavity yield an effective path length of greater than 7 km in a cell 90 cm in length. The instrument flew aboard the M55 Geophysica during the StratoClim campaign and the WB-57F during the ACCLIP campaigns. In both cases it was configured with a rear-facing inlet to make vapor phase measurements of isotopic composition. In laboratory conditions the instrument has demonstrated a measurement precision of 3.6 ppbv in $H_2O$ and 82 pptv in HDO in 5-second averages.

### 3.2 Harvard ICOS

The Harvard ICOS instrument is an OA-ICOS instrument which flew aboard the NASA WB-57F aircraft during the AVE-WIIF (2005), CR-AVE (2006), and TC4 (2007) campaigns out of Houston, Costa Rica, and Houston, respectively. The instrument uses a quantum cascade laser (QCL) to scan over $H_2O$, $H_2^{18}O$, $H_2^{17}O$, and HDO features near 6800 nm. The instrument features a 90.57 cm cell with an effective path length of about 4.5 km, and was configured with a rear-facing inlet to make vapor phase measurements of isotopic composition. Laboratory and in-flight calibrations established an accuracy of 5% for all measured species, and the instrument showed measurement precisions in 4-second averages of 0.14 ppmv, 0.10 ppbv, and 0.16 ppbv in $H_2O$, HDO and $H_2^{18}O$, respectively.

### 3.3 Hoxotope

The Hoxotope instrument made vapor phase measurements of $H_2O$ and $\delta D$ using vacuum UV photolysis of water molecules and the subsequent laser-induced fluorescence of of OH and OD fragments. This method yielded a signal-to-noise ratio of greater than 20 for 1 ppbv HDO and greater than 30 for 5 ppmv $H_2O$ in 10 second averages, sufficient for measurements of $\delta D$ in the UTLS region. The instrument flew aboard the NASA WB-57F aircraft in the AVE-WIIF and TC4 field campaigns. In the AVE-WIIF campaign, the instrument flew with a rear-facing inlet to make measurements of water vapor isotopic composition, and in the TC4 campaign it flew with a forward-facing isokinetic inlet to make measurements of total water isotopic composition. Hoxotope measurements should be particularly robust against contamination due to the instrument's high flow rate and small sample volume. Additionally, measurements made via photolysis should be more robust against contamination effects due to, e.g., outgassing within the sample cavity. Since OD and OH fragments are the actual species being measured, further outgassing of HDO and $H_2O$ downstream of the photolysis cell will not contribute to contamination.

### 3.4 ACE-FTS

The Atmospheric Chemistry Experiment (ACE) mission is a Canadian satellite mission launched on 12 August, 2003 into a high-inclination (75°) circular orbit with altitude 650 km. This orbit provides coverage from 85° S to 85° N and primarily makes observations in the middle and high latitudes. The satellite's primary instrument is a Fourier Transform Spectrometer



(FTS), which measures atmospheric absorption spectra between 2.2–13.3 μm (750–4400 cm$^{-1}$) with a resolution of 0.02 cm$^{-1}$. The instrument operates in a limb-sounding geometry, in which it observes radiation from the Sun attenuated by Earth's atmosphere at each sunrise and sunset in its orbit, of which there are about 15 of each per day. The $H_2O$ molecule is ideally sampled from 5–150 km altitude, and the HDO molecule from 5–42 or 50 km, depending on the latitude of the occultation. In practice, the lower observational limit often depends on the presence of clouds, which interfere with the volumetric mixing
ratio (VMR) retrievals. In this work, we use ACE-FTS version 5.2 retrievals (Boone et al., 2023).

The typical uncertainty associated with a single ACE-FTS profile varies with altitude and its spatial location. Figure 2 shows ACE-FTS $\delta$D retrievals from JJA 2004-2022 over the AM region and relatively cloud-free Sahara desert region. Since ACE-FTS retrievals are not generally possible through thick clouds(Boone et al., 2005), the Sahara desert region has many more retrievals down to lower altitudes than the much cloudier Asian Monsoon region. In both regions, the interquartile range (IQR)
is approximately 50 ‰ in stratospheric air above about 19.5 km. The IQR in both regions increases to a maximum at 10.5 km of about 300 ‰ in the Sahara and about 450 ‰ in the AM. The increase in IQR at lower altitudes is likely due to a combination of factors. First, the troposphere is inherently more variable than the stratosphere. Second, the presence of clouds may interfere with retrievals. Third, isotopic retrievals at low altitudes typically use weak spectral features with high temperature sensitivity. Thus, retrievals made in the troposphere, which exhibits more temperature variability than the stratosphere, may be less precise
for this reason.

Depending on the satellite's viewing angle through Earth's atmosphere, spectra are sampled with a resolution of 2-6 km, and the resulting VMRs of target species are oversampled onto an 1-kilometer grid. Similar to the ATMOS instrument, the limb sounding geometry of the ACE-FTS instrument necessarily results in sampling path lengths of approximately 200 km (RINSLAND et al., 1998).

We note here that the orientation of the SCISAT's orbit as Earth orbits the sun means that observation at a particular latitude are highly correlated with the day-of-year on which the measurement occurs, which biases the number of retrievals seasonally (cf. Figure 1 in Randel et al. (2012)). This means that for a given latitude it is not possible to construct truly seasonal averages.

## 4 Field Campaign Descriptions

We compare the ACE-FTS retrievals with in situ measurements made during five aircraft field campaigns from 2005 to 2022.
See Table 1 for number of observations ("Counts") made by each instrument during each campaign.

### 4.1 AVE-WIIF (North American Monsoon, NAM)

The Aura Validation Experiment Water Isotope Intercomparison Flight (AVE-WIIF) campaign consisted of three five-hour flights in June and July of 2005 aboard the NASA WB-57F aircraft. The flights were undertaken to compare the Hoxotope and Harvard ICOS instruments, both of which were new at the time. The flights took place out of Ellington Field (EFD) in Houston,
TX and sampled the UTLS with level legs between 10 and 19 km. During the campaign, Hoxotope returned 12.5 hours of data and Harvard ICOS returned 7.9 hours of data.



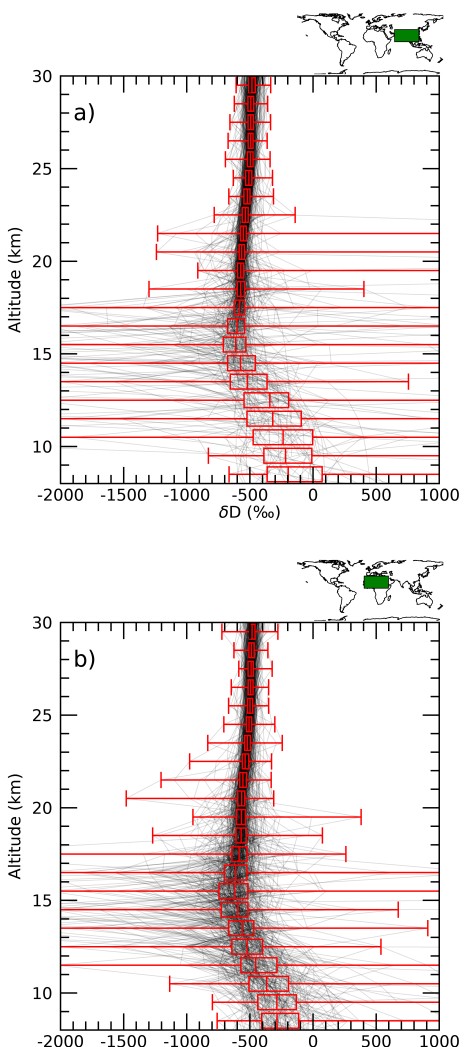

**Figure 2.** All ACE-FTS retrievals from JJA 2004-2022 for the (a) AM region and (b) relatively cloud-free Sahara desert region. The inset map in each panel shows the spatial region from which observations are taken (green boxes). Both regions have the same area and latitudinal extent (5 – 35°): AM region (60 – 120°) and Sahara region (-15 – 45°). At each altitude level, a box-and-whisker plot (cyan) shows the extrema, first quartile, median, and third quartile. The number of points considered at each altitude level varies due to the presence of clouds. In the AM region, there are 375 occultations above 20.5 km, but only 45 occultations at 8.5 km. In the less cloudy Sahara region, 350 occultations are considered above 18.5 km, and 154 at 8.5 km.

We construct ice water content (IWC) for AVE-WIIF from the total water (TW) measurements of the Harvard Total Water (HTW) instrument (Weinstock et al., 2006) and the water vapor measurements of the Harvard Water Vapor (HWV) Lyman-$\alpha$ instrument (Hintsa et al., 1999). These instruments were operational whenever HOxotope and Harvard ICOS were operational.



**Table 1.** Counts for observations between 400 K and 500 K in potential temperature. Counts represent seconds of sampling time for in situ instruments and number of occultations for ACE-FTS.

| Campaign | (Lat, Lon) | Instrument | Counts |
|---|---|---|---|
| AM | (5–35°, 60–120°) | ChiWIS | 5610 |
| | | ACE-FTS | 319 |
| AM Outflow | (10–45°, 100–160°) | ChiWIS | 54790 |
| | | ACE-FTS | 436 |
| NAM | (10–50°, 230–290°) | ChiWIS | 46535 |
| | | Harv. ICOS | 920 |
| | | Hoxotope | 9056 |
| | | ACE-FTS | 801 |
| Arctic | (40–70°, 130–240°) | ChiWIS | 42640 |
| | | ACE-FTS | 1739 |
| Tropics | (-10–20°, 260–300°) | Harv. ICOS | 9156 |
| | | ACE-FTS | 200 |

## 4.2 CR-AVE (Tropics)

The Costa Rica Aura Validation Experiment (CR-AVE) campaign took place in January and February of 2006 with 12 research flights in total. This campaign focused on providing validations of the Aura satellite, as well as the microphysical characteristics of the tropical UTLS. This campaign provided isotopic sampling of the tropics during the boreal winter months, and sampled some of the most dry and isotopically depleted air on record. During this campaign the Harvard ICOS instrument returned isotopic compositions for eight flights from the end of January to the middle of February for a total of 16.4 hours of isotopic data.

IWC in the CR-AVE campaign is provided by the NCAR counterflow virtual impactor (CVI) instrument (Twohy et al., 1997; Noone et al., 1988). The CVI inlet samples only cloud particles, evaporates them, then measures the concentration downstream with a Lyman-$\alpha$ hygrometer.

## 4.3 TC4 (Tropics)

The Tropical Composition, Cloud and Climate Coupling (TC4) campaign investigated the structure, properties, and processes in the tropical Eastern Pacific. The field campaign consisted of transit flights and research flights based in Costa Rica and Panama during boreal summer 2007. During this campaign Hoxotope returned data on 5 flights for a total of 4.6 hours of isotopic data, and Harvard ICOS on 2 flights for a total of 5.1 hours.





During this campaign the HOxotope instrument was operated with a forward-facing inlet, allowing for measurements of the total water isotopic composition. We construct an IWC measurement by subtracting the HWV water vapor values from the HOxotope total water values.

### 4.4    StratoClim (Asian Monsoon, AM)

The EU StratoClim campaign consisted of 8 flights in the Asian Summer Monsoon UTLS in July and August of 2017. The
flights took place in Kathmandu, Nepal and used the M55 Geophysica high-altitude research aircraft. The campaign aimed to produce more reliable projections of climate and stratospheric ozone by using UTLS observations of relevant trace gas species in the heart of the AM to better understand atmospheric structure in the AM anticyclone, as well as to quantify the transport of near-surface pollutants to higher altitudes. During this campaign, the instrument returned data on 6 flights for a total of 11.9 hours of isotopic data, and made measurements between 10.5–18.5 km.

The presence of clouds in StratoClim is indicated with a combination of the backscatter ratio (BR) from the Multiwavelength Aerosol Scatterometer (MAS) (Cairo et al., 2011) and the ice particle number concentration ($N_{ice}$) from the Novel Ice EXpEriment – Cloud and Aerosol Particle Spectrometer (NIXE-CAPS) instrument (Krämer et al., 2016, 2020). An interval is assessed to be cloud-free when the BR is less than 1.2 and $N_{ice} = 0$.

### 4.5    ACCLIP (NAM, Arctic, and AM Outflow)

The Asian Summer Monsoon Chemical and CLimate Impact Project (ACCLIP) field campaign aimed to investigate the transport pathways of uplifted air from within the Asian Summer Monsoon Anticyclone into the global UTLS, to sample the chemical content of AM air to better quantify AM transport, and to evaluate water transport across the tropopause to better understand the AM's role in hydrating the stratosphere. The campaign consisted of 4 test flights out of EFD in July 2021 (NAM), 3 test and research flights from EFD in July 2022 (NAM), 5 transit flights from EFD to Osan Air Base in in South Korea in
late July 2022 (Arctic), 15 research flights out of Osan in August 2022 (AM Outflow), and 4 transit flights from Osan to EFD in early September 2022 (Arctic). During this campaign, the instrument returned isotopic data on 28 out of the 31 ACCLIP flights for a total of 112.8 hours of data.

The presence of clouds in ACCLIP is indicated by the cloud flag provided by the second-generation Cloud, Aerosol, and Precipitation Spectrometer (CAPS) instrument (Dollner et al., 2023).

## 5    Methods


The in situ and remotely sensed measurements have vastly different spatial and temporal characteristics, and are not straightforward to compare. By their nature, in situ measurements are irregular, highly localized in space and time, retrieved both in and out of clouds, and oriented around local meteorology favorable to the science goals of a particular field campaign. ACE-FTS retrievals, on the other hand, are relatively regular, integrate over a large area, retrieved outside of thick clouds, and effectively
random with respect to local meteorological conditions.





ACE-FTS measurements are first filtered by removing missing or flagged retrievals as prescribed by the data usage guide. This still leaves retrievals which do not converge or have other large deviations. To avoid drawing conclusions based on these retrievals, we use the median value throughout this work when interpreting the satellite retrievals as this metric is far less susceptible to outliers in the data than the mean. For consistency, we extend this treatment to the in situ data as well.

To bridge the differences between these data sets, we first make a broad comparison between the average characteristics of the isotopic retrievals from each campaign and the characteristics of ACE-FTS VMRs for a seasonally representative latitude/-longitude box centered around the campaign region. In all cases, only cloud-free intervals from the in situ field campaigns are considered. A variety of additional measurements are used to determine the presence of clouds, which are described in each campaign subsection of Section 4.

Campaign information and parameters of the ACE-FTS averages are summarized in Table 2. This approach is most effective in cases where the sampled air masses are broadly similar over large geographic regions and/or exhibit low seasonal variations, e.g., comparisons of air in the overworld stratosphere ($\Theta > 400$ K) or within and above Earth's monsoon systems, which can seasonally dominate atmospheric composition over large regions.

## 6 Results

Figure 3 presents isotopic data for all regions considered in this study, with in situ measurements in the left column and ACE-FTS retrievals in the right column. Observations of median $\delta$D are shown in the phase-space of $H_2O$ and altitude. Viewing isotopic composition in the phase-space of water vapor variations helps distinguish/separate air masses of different origin at the same altitude level. On average, the water vapor mixing ratio is highly correlated with $\delta$D through the depth of the troposphere; it is therefore the difference in isotopic composition between airmasses with the same water vapor mixing ratio which provides insight in their convective history. See, for example, Figure 4 in Khaykin et al. (2022b) which uses this presentation to relate isotopic composition to convective influence to show the convective origin of the most isotopically enhanced airmasses.

Although this presentation of the data obscures some of the natural and instrumental variability in each bin, it still allows the identification of broad features. First, we note that in all campaign regions the ACE-FTS retrievals are isotopically lighter than the in situ observations in stratospheric air. This feature is most apparent above about 14 km in altitude and in air with 260 less than about 10 ppmv water vapor. Second, there is considerable isotopic variability between campaign regions in both the in situ measurements and the satellite retrievals.

### 6.1 Isotopic composition above 14 km

Above about 14 km, in situ isotopic measurements are often 100-200 per mil heavier than ACE-FTS retrieval. Figure 4 shows that this feature occurs across all measurement campaigns. We note here that Figure 10 of Fueglistaler et al. (2009) shows a 265 similar relationship between the ATMOS retrievals reported in Kuang et al. (2003) and ICOS measurements from the Harvard Isotope instrument. In that case, in situ measurements above about 380 K in potential temperature are approximately 150‰ enhanced with respect to the ATMOS measurements.





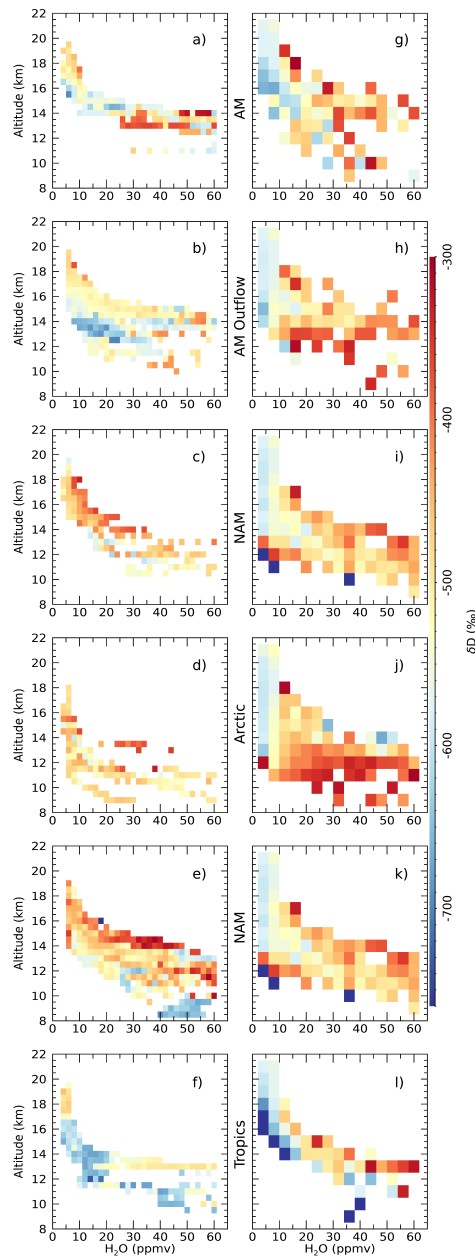

**Figure 3.** $\delta$D from in situ measurements (left col.) to satellite retrievals from ACE-FTS (right col.). Each row shows a different field campaign region. In situ observations were made by ChiWIS for (a)-(d) and Harvard ICOS/Hoxotope for (e)-(f). Data are binned by $H_2O$ and altitude at a resolution of 2 ppmv x 0.5 km for the in situ measurements and 4 ppmv x 1 km for the satellite observations. Bins are colored by their median isotopic composition.




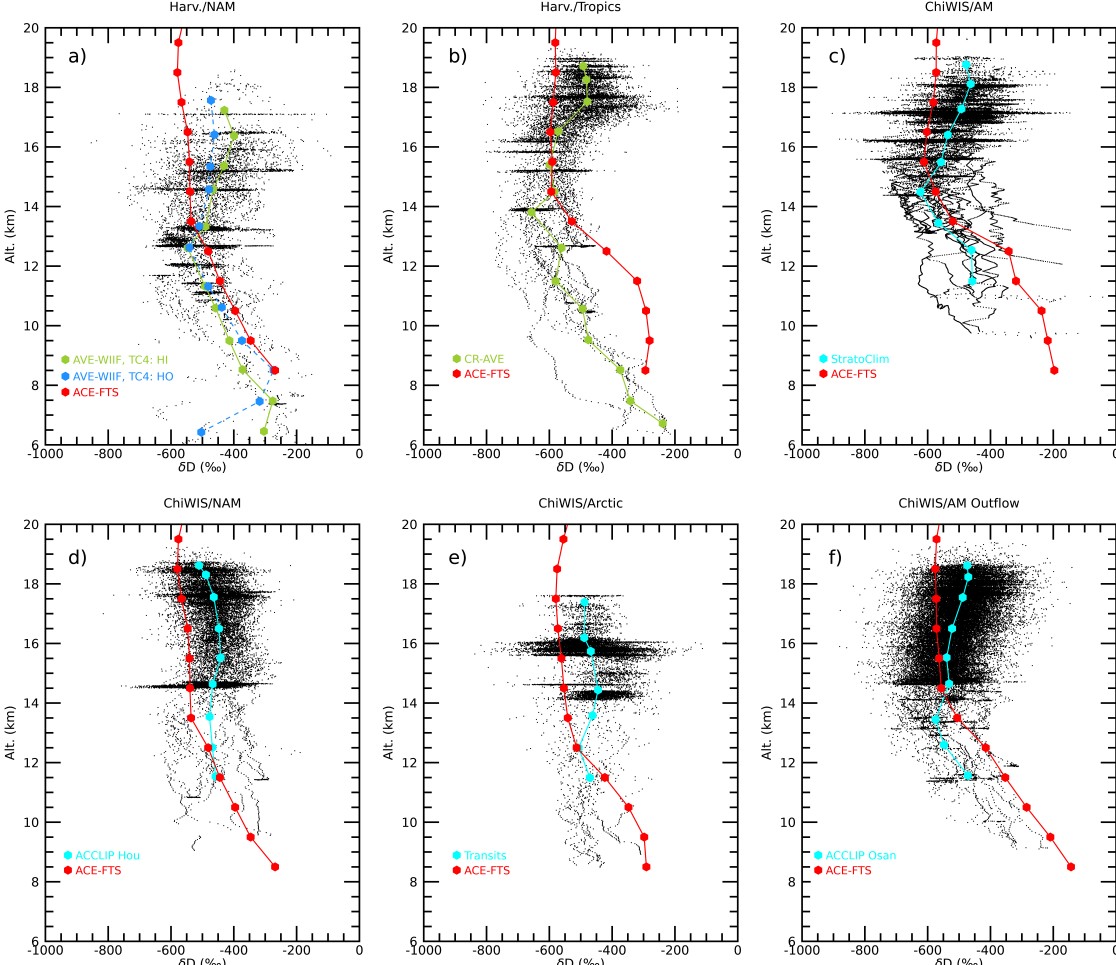

**Figure 4.** Comparison of ACE-FTS retrievals and in situ isotopic data for each campaign. In situ data are shown as black dots, and are rebinned into roughly 1-km bins. Rebinned ChiWIS data is plotted with a solid cyan line, Harvard ICOS with a solid green line, and Hoxotope with a dashed blue line. ACE-FTS median $\delta$D at each altitude level in the campaign region for the years 2004-2022 is shown with a red line.

The positive bias of the in situ measurements occurs in some of the driest regions of the UTLS ($H_2O < 4$ ppmv). Two potential causes of error in the in situ measurements may be: 1) line strength errors in the HITRAN data base (Gordon et al., 2022) or 2) contamination of the instruments' optical cavities. Regarding the first possibility, we note that ChiWIS and Harvard ICOS operate in very different wavelength regions and utilize different spectral features, thus making it unlikely that both instruments would yield the same deviations with respect to ACE-FTS retrievals at high altitudes due to line strength errors. Furthermore, during flights in which both Harvard ICOS and Hoxotope are operating, they both show enhancement above






ACE-FTS retrievals in the same region (cf. Fig. 4). The HOxotope instrument should be highly resistant to contamination due
to outgassing due to its fast response time (c.f. St. Clair et al. (2008) Figure 10) and that the LIF measurement methodology
of the Hoxotope is inherently destructive to water vapor contamination, meaning that the measurement is insensitive to $H_2O$
outgassing beyond the dissocation region. Together these suggest that HITRAN line strength errors or retrievals are responsible
for the discrepancy, although outgassing in the inlets of instruments or sampling from the aircraft's boundary layer could
contaminate observations.

Regarding the second possibility, contamination of water and isotopic retrievals, especially in dry conditions, is a serious
concern for in situ instruments. The AQUAVIT-1 water intercomparison (Fahey et al., 2014) showed at worst 20% disagreement
between water instruments between 1 and 10 ppmv, which could easily account for the increased isotopic composition of in situ
instruments on its own. However, the Harvard ICOS instrument was calibrated with two different calibration methodologies
(Sayres et al., 2009), and ChiWIS showed excellent agreement with two other in situ water vapor measurements during the
StratoClim campaign (Singer et al., 2022).

Comparisons of $H_2O$ and HDO observations in stratospheric air are presented in Figure 5. The histograms of each data
set are normalized to ease comparison of their means and standard deviations. The statistical characteristics of $H_2O$ and
HDO observations for each data set are summarized in Table 2 and Table 3, respectively. Table 1 summarizes the sampling
information including spatiotemporal location of each region and instrument sampling counts. The instrument counts represent
the number of seconds of observations for in situ measurements, and the number of occultations for ACE. We note that the
ACE-FTS retrievals are far more uniform than those from in situ campaigns, which likely reflects fine scale structure which is
averaged out in the ACE-FTS retrievals.

In situ observations of water vapor (Fig. 5a) show considerable variability between in situ field campaigns. Harvard ICOS
measurements in the CR-AVE campaign, and ChiWIS measurements in the NAM and Arctic have modes at or just above
4 ppmv, although in each case the distributions show skew towards higher mixing ratios. This skew may be due to deliberate
targeting of convective outflow during these campaigns, resulting in oversampling of wet, isotopically enriched air. Campaigns
associated with the AM region (StratoClim and ACCLIP measurements of AM outflow) show broader distributions, which
likely reflect the export of moisture to high altitudes by convection in the AM system. $H_2O$ measurements from ACE-FTS
(Fig. 5b) are very consistent across all measurement regions, with modal values between 3.75 and 4 ppmv. These observations
likely show less variability than in situ observations in large part due to their large spatial averaging windows. The long
climatological interval over which the ACE-FTS measurements are averaged may also serve to wash out annual variations in
the lower stratosphere.

In situ observations of HDO are inherently noisier than $H_2O$ observations, which therefore masks some natural variability.
These observations have more normal distributions but generally cluster in ways similar to the $H_2O$ observations (Fig. 5c). The
Harvard ICOS measurements in the CR-AVE campaign, and ChiWIS measurements in the NAM and Arctic all have modal
values around 0.65 ppbv. Campaigns associated with the AM region have distributions of similar width, but with modal values
around 0.80 ppbv. The large spreads in HDO retrievals observed in ChiWIS/AM outflow, ChiWIS/AM, Harvard ICOS/NAM,
and HOxotope/NAM data could be the result of several different effects. First, it is possible the spread is a consequence of



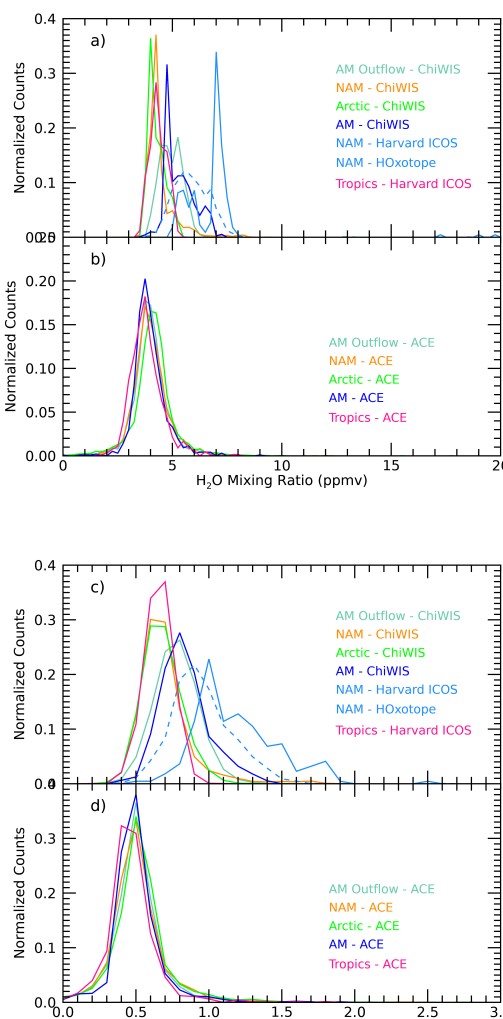

**Figure 5.** Normalized probability density functions of (a-b) $H_2O$ and (c-d) HDO comparing in situ measurements (a,c) and satellite retrievals (b,d) in stratospheric conditions. Stratospheric air is defined here as conditions with potential temperature between 400 K and 500 K. ACE-FTS retrievals over regions aligned with each corresponding aircraft campaign (color coded) located in the campaign bounding boxes shown in Figure 1 and defined Table 1, and are drawn from the appropriate season over the entire 2004-2022 interval.

some form of contamination. Second, the spread could be the result of biases introduced by the scientific objectives of the
310  campaign, e.g., flying in and out of convective plumes would likely manifest as broad variability in these plots. As with $H_2O$,
ACE-FTS HDO retrievals are nearly identical throughout all measurement regions (Fig. 5d).



**Table 2.** Mean, standard deviation, and median for $H_2O$ observations between 400 K and 500 K in potential temperature. Counts represent seconds of sampling time for in situ instruments and number of occultations for ACE-FTS. All units are in ppmv.

| Campaign | Instrument | Mean | SD | Median |
|----------|-----------|------|------|--------|
| AM | ChiWIS | 5.46 | 0.67 | 5.32 |
| | ACE-FTS | 4.09 | 0.78 | 4.00 |
| AM Outflow | Chiwis | 5.06 | 0.50 | 5.04 |
| | ACE-FTS | 4.11 | 0.81 | 4.07 |
| NAM | Chiwis | 4.53 | 0.69 | 4.35 |
| | Harv. ICOS | 8.22 | 6.86 | 7.10 |
| | Hoxotope | 6.03 | 1.16 | 5.94 |
| | ACE-FTS | 4.11 | 0.87 | 4.05 |
| Arctic | ChiWIS | 4.40 | 0.39 | 4.31 |
| | ACE-FTS | 4.21 | 1.13 | 4.23 |
| Tropics | Harv. ICOS | 4.47 | 0.38 | 4.44 |
| | ACE-FTS | 3.94 | 0.74 | 3.88 |

Significant interannual variability in both $H_2O$ and HDO has been observed in the UT/LS, which potentially account for some of the difference observed between in situ and remote sensing measurements. Randel et al. (2012) observe the tropical tape recorder signal in ACE-FTS retrievals to have variations in both species of about 25%, and of about 50 ‰ in $\delta D$. They observe similar isotopic variability in the summer North American and Asian monsoon systems. Clouser et al. (2024) also observe similar isotopic variability in high-altitude observations during flights out of Houston in the NAM during the years 2021 and 2022.

To assess this variability, we calculate the annual means in $H_2O$, HDO, and $\delta D$ using the ACE-FTS data set for 2006-2022 for each observation region in this study. The results are summarized in Table 4.

Cross-referencing the instrumental means for each campaign from Table 2 and Table 3 with the means and standard deviations in Table 4 shows that interannual variability is unlikely to be responsible for the observed differences, assuming that the interannual variability of the satellite retrievals is representative of what would be measured in situ. In the case of $H_2O$ and HDO, the mean of each set of campaign measurements is two or more standard deviations wetter than the mean of the annual ACE-FTS retrievals, with the exception of the high-latitude measurements made by ChiWIS during its transits between Osan AB and Houston.

In general, ACE-FTS retrievals of $H_2O$ are slightly drier than those of in situ campaigns. HDO satellite retrievals, however, are significantly lower than in situ measurements. We therefore conclude that the HDO retrievals are the main driver of the difference between in situ and satellite retrievals of $\delta D$. This observation, combined with the long sampling path length of the ACE-FTS instrument, also suggests the possibility of condensation impacting one of the observations more than the



**Table 3.** Mean, standard deviation, and median for HDO observations between 400 K and 500 K in potential temperature. Counts represent seconds of sampling time for in situ instruments and number of occultations for ACE-FTS. All units are in ppbv.

| Campaign | Instrument | Mean | SD | Median |
|----------|-----------|------|-----|--------|
| AM | ChiWIS | 0.88 | 0.17 | 0.86 |
| | ACE-FTS | 0.55 | 0.18 | 0.53 |
| AM Outflow | ChiWIS | 0.83 | 0.15 | 0.82 |
| | ACE-FTS | 0.55 | 0.20 | 0.54 |
| NAM | ChiWIS | 0.75 | 0.19 | 0.72 |
| | Harv. ICOS | 1.55 | 1.52 | 1.21 |
| | Hoxotope | 0.99 | 0.42 | 0.98 |
| | ACE-FTS | 0.55 | 0.22 | 0.54 |
| Arctic | ChiWIS | 0.73 | 0.13 | 0.72 |
| | ACE-FTS | 0.57 | 0.26 | 0.57 |
| Tropics | Harv. ICOS | 0.71 | 0.10 | 0.71 |
| | ACE-FTS | 0.51 | 0.18 | 0.50 |

**Table 4.** Mean of annual means and standard deviation of annual means for ACE-FTS retrievals of $H_2O$, HDO, and $\delta D$ between 400 K and 500 K in potential temperature. Counts represent seconds of sampling time for in situ instruments and number of occultations for ACE-FTS. $H_2O$ units are ppmv, HDO units are ppbv, and $\delta D$ units are ‰. Individual yearly averages are indicated by a single overbar. The expectation value of each yearly average is indicated with the $E$ symbol, and is roughly equivalent to the averages presented in Tables 2 and 3.

| Region | $E(\overline{H_2O})$ | $\sigma(\overline{H_2O})$ | $E(\overline{HDO})$ | $\sigma(\overline{HDO})$ | $E(\overline{\delta D})$ | $\sigma(\overline{\delta D})$ |
|--------|------|------|------|------|------|------|
| AM | 4.13 | 0.22 | 0.56 | 0.04 | -556 | 53 |
| AMO | 4.12 | 0.16 | 0.55 | 0.02 | -578 | 15 |
| NAM | 4.13 | 0.18 | 0.55 | 0.03 | -575 | 11 |
| Arctic | 4.22 | 0.20 | 0.57 | 0.04 | -576 | 18 |
| Tropics | 3.98 | 0.22 | 0.52 | 0.04 | -585 | 13 |

330 other. Lower $H_2O$ and significantly lower HDO are characteristic of a low-temperature isotopic fractionation effect. The geometry of ACE-FTS measurements makes it more susceptible to condensation effects than in situ measurements due to its long observational path length (see Section 3.4), and ability to sample through thin cirrus (Eremenko et al., 2005). This could also explain the similar discrepancy in average observed for the previous study's comparison with ATMOS results (Section 6.1), since ATMOS had essentially the same measurement geometry as ACE. Even if both the in situ and ACE-FTS measurements

335 are unbiased, it is still possible the instruments might return significantly different isotopic compositions from the same general region due to their very different sampling methodologies.





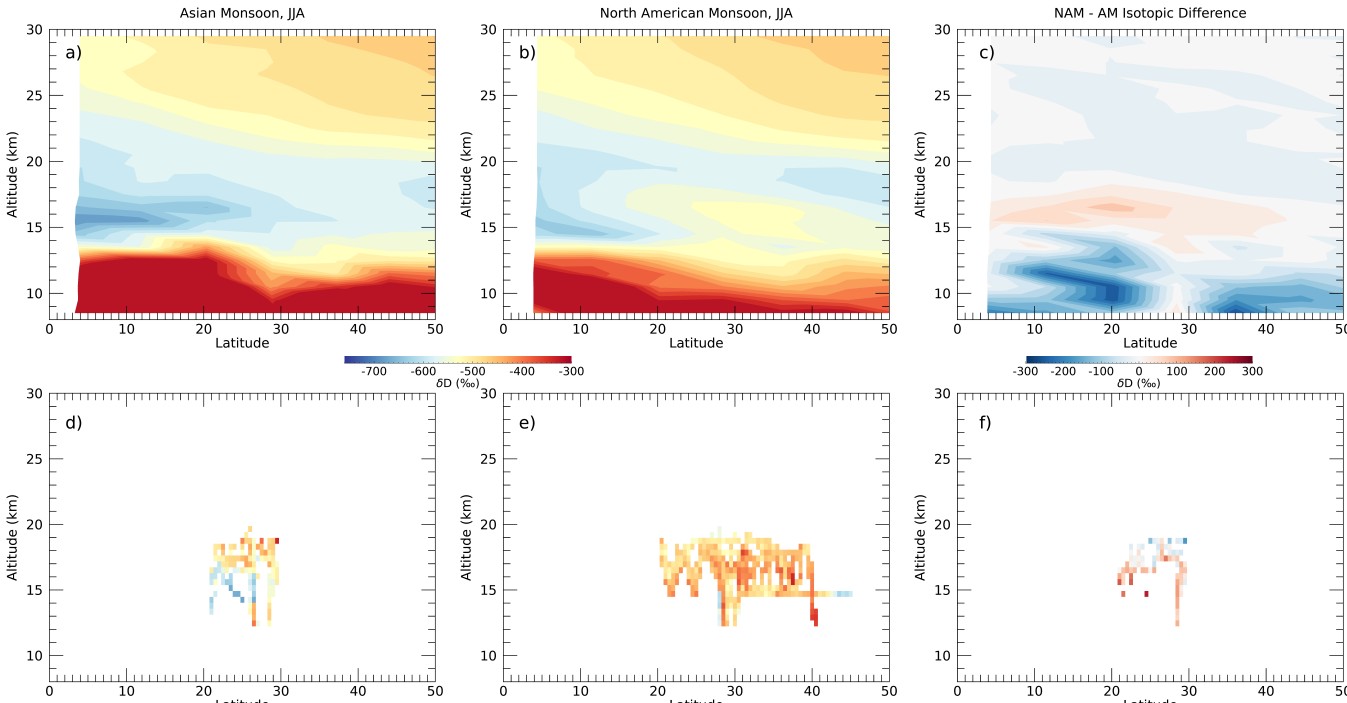

**Figure 6.** ACE-FTS (top row) and ChiWIS (bottom row) isotopic compositions for the Asian Monsoon (left), North American Monsoon (middle), and their difference (right). The ACE-FTS data in each longitudinal wedge between the latitudes of 0° and 50° N and altitudes of 8–30 km are rebinned into 6.25° by 1 km boxes, and the isotopic compositions shown are the median value of each bin. The Asian Monsoon region is defined here to be between 60° and 120° longitude, and the North American Monsoon region to be between 230° and 290°. A similar procedure is followed with the ChiWIS in situ data, although the boxes are 0.9° by 0.5 km. The difference in isotopic composition between boxes is calculated only for boxes in which ChiWIS made measurements in both the NAM and AM.

## 6.2 Isotopic Enhancement above the North American Monsoon

In the region of common measurements between 19° and 30° latitude and 15 km and 19 km altitude, the ACE-FTS and ChiWIS observations both show enhancement above the NAM, with values of 46 ‰ and 33 ‰, respectively. Figure 6 shows this isotopic composition in the AM and NAM regions for ACE-FTS retrievals (top row) and ChiWIS in situ measurements (bottom row). The isotopic enhancement over the NAM region was first noted in Randel et al. (2012), who attributed the differences in isotopic composition to the background thermodynamic structure and differences in relative humidity. Here we make use of the increased number of ACE-FTS observations to construct meridional plots of the isotopic structure over the Asian and North American Monsoon regions.



345 In the overworld stratosphere (above about 18 km), ACE-FTS retrievals show evidence of increasing $\delta$D due to methane oxidization, although as expected the difference plot shows little difference in this altitude range between the two regions. High-altitude research aircraft do not reach altitudes where methane oxidization is a significant effect.

 In the transition region between about 15–18 km, satellite retrievals show the NAM region to be significantly enriched compared to the AM region, confirming that this result is still present in the v5.2 retrievals. Interestingly, the point at which

350 the NAM is most enhanced (approximately 35° N and 15.5 km) does not correspond to the point with the largest difference between the NAM and AM regions, which occurs at approximately 20° N and 16.5 km. These differences call for a more detailed investigation into their origins.

 Below 15 km, the AM region is much more isotopically enhanced than the NAM region, primarily because it is much wetter, and wetter air tends to be more isotopically enriched.

## 355 7 Conclusions

Here we present the first systematic comparison of water vapor isotopic composition from satellite and in situ retrievals. This work spans five measurement campaigns, covers the northern hemisphere from approximately 10° N to 60°, and the Asian and North American Monsoon systems. The field campaigns span the years from 2006 to 2022, providing significant overlap with the operational years of the ACE-FTS instrument.

360 This work compares the $H_2O$, HDO, and $\delta$D data sets in three ways. Climatological averages of $\delta$D are compared to in situ measurements to look for systematic deviations between the data sets in terms of systematic biases over certain altitude intervals, inconsistent measurement envelopes, and large regional differences. This qualitative comparison shows that the in situ retrievals of $\delta$D in the lower stratosphere are consistently higher than those of the ACE-FTS instrument.

 Detailed investigation of the in situ measurements shows that they are consistently about 100‰ more isotopically enriched

365 than the median ACE-FTS $\delta$D retrievals above 14 km in the same region. This difference holds across measurements made by the ChiWIS, Harvard ICOS, and Hoxotope instruments, the last of which should be especially resistant to contamination due to its measurement principle. We take this to be evidence that the spectroscopic features used in the satellite retrievals have error in high-altitude retrievals, and note that while both $H_2O$ and HDO satellite retrievals are biased low, the HDO retrievals are significantly more so. However, we cannot fully rule out bias in the in situ instruments due to contamination. In any case,

370 resolving these large differences (>100‰) is important as the limited in situ and remote sensing measurements of $\delta$D at these altitudes form the only basis for constraining isotopically-enabled GCMs.

 It is noteworthy that the satellite/in situ difference is smallest in both species in the high-latitude transits of ChiWIS during the ACCLIP campaign, and largest in the CR-AVE tropical measurements. Three possible causes present themselves: a) the high-latitude flights have no science targets and no clear bias towards a particular type of airmass, b) the 400 K to 500 K

375 potential temperature range is found at a lower altitude in the high-latitudes, and retrievals there are therefore spread across a different set of microwindows than those in the tropics and mid-latitudes, and c) the ACE-FTS instrument observes cirrus clouds over a relatively long path length in the tropics resulting in depletion in $\delta$D relative to localized in situ measurements.





In situ measurements of $\delta$D in both the NAM and AM by the ChiWIS instrument confirm the isotopic enhancement over the NAM reported in ACE-FTS observations by Randel et al. (2012). These differences likely reflect the specific thermodynamic

and relative humidity structure of the NAM and AM systems. Further investigation is needed to fully exploit the information contained in these isotopic measurements.

Observations of water isotopologues in Earth's upper atmosphere are a powerful tool for understanding the influence of convection and transport of moisture into the region. Fundamental spectroscopy is needed to improve satellite retrievals of $\delta$D, which could then be more effectively used to constrain the global the water vapor budget of the TTL and isotopically

enabled GCMs. Furthermore, increased sampling frequency is needed in both in situ measurements and in the next generation of remote sensing platforms. Ideally, this sampling would comprise a research payload targeted to ACE-FTS measurements aboard a high-altitude aircraft such as the WB-57F or ER-2, with flight paths co-located with occultations.

*Code and data availability.* ChiWIS data from the 2017 StratoClim campaign is be accessible via the HALO database at https://halo-db.pa.

op.dlr.de/mission/101. Data from the ACCLIP campaign is accessible via the NASA LaRC database at https://www-air.larc.nasa.gov/cgi-bin/

ArcView/acclip. Data from the CR-AVE, TC4, and AVE-WIIF campaigns are available on the NASA ESPO archive at https://espoarchive.

nasa.gov/archive/browse. ACE-FTS v5.2 retrievals are available at https://databace.scisat.ca. The software used to process, analyze, and

visualize the ACE-FTS data can be found at https://github.com/bwclouser/ACE-FTS-Handler.

*Author contributions.* BWC conceived of the study and performed the analysis. BWC and CES wrote the manuscript, with contributions

from all coauthors. AD, CCK, JMS, TFH, RSS, and EJM provided feedback on the manuscript.

*Competing interests.* One of the co-authors is a member of the AMT editorial board.

*Acknowledgements.* BWC and CCK acknowledge support by the NSF through the Partnerships in International Research and Education

(PIRE) program (Grant No. OISE-1743753). We thank the StratoClim coordination team, Myasishchev Design Bureau, and Geophysica

pilots and ground crew for the successful execution of the StratoClim campaign. We thank the NASA support team and the WB-57 pilots

and ground crew for the successful execution of the ACCLIP campaign.

*Financial support.* This work was supported by the National Science Foundation through the Partnerships in International Research and

Education program under grant number OISE-1743753. This research has been supported by the StratoClim project of the European Com-

munity's Seventh Framework Programme (FP7/2007–2013) under grant agreement no. 603557. The ACCLIP campaign was supported by

NSF, NASA, and NOAA.



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
