# Peer review of "A systematic comparison of ACE-FTS $\delta D$ retrievals with airborne in situ sampling"

_EGUsphere, 2025_

## Referee Comment (RC1)

This article collects the results of several in situ H2O/HDO campaigns and compares them to observations from the ACE-FTS solar occultation satellite mission. This provides an opportunity to place the in situ measurements into a broader context using the global coverage of ACE-FTS measurements. It also implicitly serves as a cross-validation of the different in situ measurements, if they can be shown to have similar systematic biases compared to the ACE-FTS results.

The paper is well written and seems reasonably complete. I only have a few comments, the most significant being an underemphasis of evidence of a systematic bias between the data sets. I should point out that this underemphasis appears to be a consequence of feedback that I, myself, provided the lead author in the past.

HDO enhancement in the North American monsoon is an interesting puzzle.

Major comments

> In Table 1, it is not clear how the ACE occultations are selected in order to get the reported numbers of occultations. I assume the selection took all occultations in the latitude and longitude ranges for all years from 2004 to 2022, but only for the same month(s) as the corresponding campaign? It should probably be mentioned in the text.

> Figure 4: perhaps include a dotted line in each panel showing an estimate of the tropopause height (from MERRA-2 or some other source). It is a significant consideration for interpreting the results. For example, the results near 16 km in the tropical measurements are in the troposphere, while in the Arctic, the results near 16 km are in the stratosphere. It might also be instructive to show the average altitude corresponding to the 400 K potential temperature level.

> Line 290: We note that the ACE-FTS retrievals are far more uniform than those from in situ campaigns, which likely reflects fine scale structure which is averaged out in the ACE-FTS retrievals.

This actually seems to be more of a sampling issue. The two sampling quantities (seconds for the in situ measurements versus individual measurements for ACE) are not entirely compatible. Shown below is an excerpt of Figure 5. Consider the curves for "NAM – Harvard ICOS" and "NAM – HOxotope." On a side note, I cannot tell which curve corresponds to the dashed line and which one corresponds to the solid line.

I assume these NAM curves are derived from a limited number of flights that targeted relatively moist events. If the plane is essentially "wallowing" in moist air, the distribution will of course skew to higher mixing ratio, but taking each second of measurement as a separate count is effectively counting the same "moist events" multiple times (many, many times). Also shown below is an excerpt from Figure 3, with the ACE results for North American monsoon contained in panel k. Note that at 16-17 km, ACE saw H2O mixing ratios up to 20 ppm, but the fraction of moist events in that altitude region seen was small enough that it doesn't visually register in the probability distribution in Figure 5. ACE results agree that higher H2O concentrations occur, but not as frequently as the in situ plots might imply.

In summary, the fraction of seconds you measure high H2O mixing ratios when sitting in moist air is not equivalent to the fraction of times you measure high H2O mixing ratios when randomly sampling a particular geographic region with ACE.

[Figure]

Above: excerpt from Figure 5

[Figure]

Above: excerpt from Figure 3

> Line 326: In general, ACE-FTS retrievals of H2O are slightly drier than those of in situ campaigns.

Figure 5 (and the discussion in the text) gives the worrisome impression that the in situ and ACE results are inconsistent, but I believe that is somewhat misleading. Consider the H2O distribution for the Arctic case in Figure 5. This is the only set for which 400 to 500 K potential temperature would be well into the stratosphere for all the measurements. The in situ measurements would presumably be random samples of the stratosphere along the route (rather than targeting a local moist air event), more similar in nature to the ACE measurements. For the Arctic, the probability distributions for ChiWIS and ACE appear to overlap fairly well, suggesting reasonable agreement between the two data sets.

For the other in situ cases, the 400 to 500 K potential temperature range includes a portion of the upper troposphere, and the in situ study presumably targets a most air event, leaving the impression that ACE is usually biased dry, but again I expect that is an artifact of the sampling. If one were performing validation or calibration, the pure stratospheric measurements in the Arctic would be the preferred data set for comparison. There are fewer complicating factors.

HDO, on the other hand, does not overlap particularly well for ACE and the in situ measurements, not even for the Arctic case. This is strong evidence of an inconsistency in the HDO results, but it is difficult to say which side has the problem (or perhaps it is both sides). From the ACE perspective, the likely source of a bias would be errors in the spectroscopy (i.e., line intensities). Because different lines are employed in different altitude regions, this bias could have an altitude dependence.

> Line 333: Even if both the in situ and ACE-FTS measurements are unbiased, it is still possible the instruments might return significantly different isotopic compositions from the same general region due to their very different sampling methodologies.

This is the subject I previously discussed with the lead author. There are instances in the comparison set that seem indicative of clouds impacting the results (through enhanced condensation of atmospheric HDO relative to main isotopologue H2O). For example, in the excerpt from Figure 3 below, showing the results from measurements in the tropics, the magnitude of the apparent fractionation in the 15 to 18 km altitude range is very large in the ACE results. In the tropics, this altitude range is in the troposphere, and ACE observes cirrus clouds relatively frequently at these altitudes in the tropics, frequently enough for lower levels of HDO to be seen in the average results. These results might be improved by using the ACE 1 micron imager data to identify occultations containing cirrus clouds and excluding them from the analysis, although I am not suggesting such filtering is necessarily required in revisions to this paper. It is simply worth keeping in mind that the larger discrepancies in the tropics have a logical explanation.

[Figure]

However, there is no similar argument available to explain differences in comparisons in the Arctic set, where 16 to 18 km is in the lower stratosphere (not the troposphere). During the summer (the time frame of the in situ Arctic measurements), there should be no clouds in the stratosphere for latitudes of 40 to 70 °N, so no possibility of them impacting the HDO fractionation. Over the years, there have been periodic volcanic eruptions that provided enhancements in stratospheric sulfate aerosols (liquid droplets of H2O and H2SO4 mixtures), which could provide a means of preferentially reducing atmospheric HDO through condensation, but volcanic eruptions that dramatically enhance stratospheric sulfates are not so frequent that they should pull the average significantly.

Looking at the excerpt of Figure 4 shown below, containing the Arctic results, the ~100 ‰ offset in the stratosphere cannot be explained away by clouds impacting the ACE results. Again, the most reasonable explanation would be that there is a systematic bias between the two instruments, some aspect in the analysis of one or both data sets that induces a systematic error in HDO fractionation, likely in the determination of HDO mixing ratio itself (rather than main isotopologue H2O). In my opinion, the strongest argument for the bias can be made by looking at pure stratospheric measurements in the Arctic set, but we also see similar offsets near 18 km for the other data sets in Figure 4, a persistent offset that strongly suggests issues in one or both data sets. Problems in the HDO line strengths wouldn't shock me, but there is no way to know for certain which data set is the biggest culprit in creating the bias.

Again, it would be appropriate to put stronger emphasis in the text on this systematic difference. Apologies if my previous feedback made the authors hesitant to highlight the discrepancy.

[Figure]

ChiWIS/Arctic

Minor comments

> Line 66: The instrument has been in operation from 2005 through the present day

   The ACE-FTS has been in operation since 2004.

> Line 96: HOxotope

   This instrument name is sometimes written as "Hoxotope" and other times as "HOxotope."

> Line 153: high-inclination (75◦) circular orbit

   The inclination of the orbit is 74 degrees

> Line 157: The $H_2O$ molecule is ideally sampled from 5–150 km altitude

   ACE-FTS $H_2O$ retrievals extend up to 95 km.

> Line 174: (RINSLAND et al., 1998)

   All capital letters

> Index to Table 2 (and Tables 3 and 4): Counts represent seconds of sampling time for in situ instruments and number of occultations for ACE-FTS

   There are no counts in these tables.  There were counts in Table 1, so this is clearly a copy-and-paste issue.

---

## Community Comment (CC2)

Hello Dr. Khosrawi,

Thanks for these comments, they are highly appreciated.

Regarding the ACE team, their policy, communicated to me in an email, is this: "If one of our team members were involved in data analysis then a co-authorship would be warranted. Otherwise, we don't ask for co-authorship but only acknowledgement and reference." However, I'll ask again for good measure.

Thank you for the references. I am unfamiliar with several of the newer ones.

>A major point that does not become clear from your study is if you are assessing the accuracy of the
>ACE-FTS isotope measurements or the accuracy airborne in-situ instruments? What actually is the
>intention of the study should be more clearly pointed out.

This is a good point, and in revising the paper I will make it more clear what the main point is. The original intent with this study was to respond to statements in the literature such as those found in Fueglistaler et al. 2009 (see Section 2.6 and Figure 10) suggesting a need for reconciliation between satellite and in situ measurements of water isotopologues before either could properly be used for interpretation.  As with so many things, it is not that simple. As Chris notes in his review either or both of the satellite and in situ retrievals could be biased, and the biases may change based on altitude, the presence of cirrus, or the sampling priorities of the in situ measurements.

Thanks again,

Ben

---

## Author Comment (AC1)

The authors thank the review for the comments below. Replies to the comments are in red.

The paper illustrates a series of relevant results, tackling the problem of comparing heterogeneous in situ measurements of D/H with ACE-FTS satellite data products. This itself is a complicated problem, and the paper serves as an effort to cross-validate several datasets, illustrating consistencies and potential issues.

I only have a few, mostly minor comments that the authors should address, and which are reported here below:

1) In the introduction, the authors should also consider adding results from nadir hyperspectral sensors such as IASI, which allow for HDO and D/H vertical profile retrieval on a global scale. Specifically the following works:

https://acp.copernicus.org/articles/9/9433/2009/

https://essd.copernicus.org/articles/14/709/2022/

https://www.sciencedirect.com/science/article/pii/S0022407316301248?via%3Dihub

provide a good overview of results whoch would be beneficial to this introductory discussion.

The authors have added a section of text in the introduction describing these measurements.

2) Line 156: to be more precise, and to distinguish ACE-FTS from instruments like MIPAS, "limb-sounding" should be replaced by "solar occultation"

"Limb-sounding" has been replaced by "solar occultation" in the text.

3) Figure 4: It would be useful to report the variability of ACE-FTS profiles averaged here with errorbars.

The authors have included error bars representing the inter-quartile range of the ACE profiles to Figure 4, and report them in the text.

4) Lines 270 - 272: this is one of the core points of the paper. I wonder if the authors checked the possibility that temperature dependency of line strengths in different spectral regions cause this. This would clearly be another factor beside the already noted difference in sampling techniques between ACE-FTS and in situ techniques.

The authors believe this comment refers to the possibility that the relationship between temperature and altitude in the atmosphere could result in the general divergence between ACE and in situ measurements which occurs around 12-14 km throughout the measurement regions. That is, there may be a temperature-dependent bias in the in situ measurements. However, both ChiWIS and Harvard ICOS are extractive instruments and maintain a nearly constant optical bench temperature at about standard room temperature throughout sampling. Thus while there may be some temperature dependent

effect on the line strength, the sample gas temperature is independent of altitude so such an effect would be constant in altitude. This important point has been incorporated into the text.

5) Lines 309 - 311: to better discuss this, the manuscript should report the exact space and time box considered for ACE-FTS, and eventually include comparison with a restricted dataset of ACE-FTS profiles.

The space time boxes are now clearly denoted in the text upon the first occurrence of each measurement region, and a column has been added to Table 1 denoting the time interval of the measurements.

Additionally, the authors have expanded the methods section to make clear how exactly the ACE and in-situ measurements are being compared, and that the comparisons presented are being made on restricted data sets of ACE profiles, a point which was not clear in previous versions of the text.

6) Sometimes UT/LS is used, some other times UTLS. Please make the notation uniform across the paper.

We have changed UT/LS to UTLS throughout.

---

## Author Comment (AC2)

The authors thank the reviewer for the comments below. Replies to the comments are in red.

This article collects the results of several in situ $H_2O$/HDO campaigns and compares them to observations from the ACE-FTS solar occultation satellite mission. This provides an opportunity to place the in situ measurements into a broader context using the global coverage of ACE-FTS measurements. It also implicitly serves as a cross-validation of the different in situ measurements, if they can be shown to have similar systematic biases compared to the ACE-FTS results.

The paper is well written and seems reasonably complete. I only have a few comments, the most significant being an underemphasis of evidence of a systematic bias between the data sets. I should point out that this underemphasis appears to be a consequence of feedback that I, myself, provided the lead author in the past.

HDO enhancement in the North American monsoon is an interesting puzzle.

Major comments

> In Table 1, it is not clear how the ACE occultations are selected in order to get the reported numbers of occultations. I assume the selection took all occultations in the latitude and longitude ranges for all years from 2004 to 2022, but only for the same month(s) as the corresponding campaign? It should probably be mentioned in the text.

The selection criteria for ACE occultations over each selected measurement region are now detailed in the text, and a column added to Table 1 indicating the selection time interval. Indeed, the temporal component is centered on the relevant campaign, and averaged over all years up to 2022.

> Figure 4: perhaps include a dotted line in each panel showing an estimate of the tropopause height (from MERRA-2 or some other source). It is a significant consideration for interpreting the results. For example, the results near 16 km in the tropical measurements are in the troposphere, while in the Arctic, the results near 16 km are in the stratosphere. It might also be instructive to show the average altitude corresponding to the 400 K potential temperature level.

The authors have added a line indicating the tropopause height to each of the sub-panels. Discussion is included in the text as well.

> Line 290: We note that the ACE-FTS retrievals are far more uniform than those from in situ campaigns,which likely reflects fine scale structure which is averaged out in the ACE-FTS retrievals.

This actually seems to be more of a sampling issue. The two sampling quantities (seconds for the in situ measurements versus individual measurements for ACE) are not entirely compatible. Shown below is an excerpt of Figure 5. Consider the curves for "NAM – Harvard ICOS" and "NAM – HOxotope." On a side note, I cannot tell which curve corresponds to the dashed line and which one corresponds to the solid line. (Added better indicators in the plot to distinguish these curves.)

I assume these NAM curves are derived from a limited number of flights that targeted relatively moist events. If the plane is essentially "wallowing" in moist air, the distribution will of course skew to higher mixing ratio, but taking each second of measurement as a separate count is effectively counting

the same "moist events" multiple times (many, many times). Also shown below is an excerpt from Figure 3, with the ACE results for North American monsoon contained in panel k. Note that at 16-17 km, ACE saw H2O mixing ratios up to 20 ppm, but the fraction of moist events in that altitude region seen was small enough that it doesn't visually register in the probability distribution in Figure 5. ACE results agree that higher H2O concentrations occur, but not as frequently as the in situ plots might imply.

In summary, the fraction of seconds you measure high H2O mixing ratios when sitting in moist air is not equivalent to the fraction of times you measure high H2O mixing ratios when randomly sampling a particular geographic region with ACE.

Indeed, comparison is especially difficult in more heterogeneous regions of the UTLS. The authors have reworked this text to avoid drawing or implying general conclusions, and to focus more on the difficulties of rigorous intercomparison in heterogeneous air. This also nicely sets up the subsequent discussion of the 'good' sampling to be had in stratospheric, arctic air in which there is no distinct science goal to bias the measurements, and essentially no clouds.

> Line 326: In general, ACE-FTS retrievals of H2O are slightly drier than those of in situ campaigns.

Figure 5 (and the discussion in the text) gives the worrisome impression that the in situ and ACE results are inconsistent, but I believe that is somewhat misleading. Consider the H2O distribution for the Arctic case in Figure 5. This is the only set for which 400 to 500 K potential temperature would be well into the stratosphere for all the measurements. The in situ measurements would presumably be random samples of the stratosphere along the route (rather than targeting a local moist air event), more similar in nature to the ACE measurements. For the Arctic, the probability distributions for ChiWIS and ACE appear to overlap fairly well, suggesting reasonable agreement between the two data sets.

For the other in situ cases, the 400 to 500 K potential temperature range includes a portion of the upper troposphere, and the in situ study presumably targets a most air event, leaving the impression that ACE is usually biased dry, but again I expect that is an artifact of the sampling. If one were performing validation or calibration, the pure stratospheric measurements in the Arctic would be the preferred data set for comparison. There are fewer complicating factors.

Agreed.  The authors now center the discussion here more on the sampling difficulties introduced by in situ measurement bias, and draw a clearer distinction between the Arctic case and the other cases. We note that the H2O distributions in the Arctic overlap relatively well.

HDO, on the other hand, does not overlap particularly well for ACE and the in situ measurements, not even for the Arctic case. This is strong evidence of an inconsistency in the HDO results, but it is difficultto say which side has the problem (or perhaps it is both sides). From the ACE perspective, the likely source of a bias would be errors in the spectroscopy (i.e., line intensities). Because different lines are employed in different altitude regions, this bias could have an altitude dependence.

> Line 333: Even if both the in situ and ACE-FTS measurements are unbiased, it is still possible the instruments might return significantly different isotopic compositions from the same general region due to their very different sampling methodologies.

This is the subject I previously discussed with the lead author. There are instances in the comparison set that seem indicative of clouds impacting the results (through enhanced condensation of

atmospheric HDO relative to main isotopologue H2O). For example, in the excerpt from Figure 3 below, showing the results from measurements in the tropics, the magnitude of the apparent fractionation in the 15 to 18 km altitude range is very large in the ACE results. In the tropics, this altitude range is in the troposphere, and ACE observes cirrus clouds relatively frequently at these altitudes in the tropics, frequently enough for lower levels of HDO to be seen in the average results. These results might be improved by using the ACE 1 micron imager data to identify occultations containing cirrus clouds and excluding them from the analysis, although I am not suggesting such filtering is necessarily required in revisions to this paper. It is simply worth keeping in mind that the larger discrepancies in the tropics have a logical explanation. (This would indeed be an interesting study which we hope to pursue, but feel that it is beyond the scope of the current paper.)

However, there is no similar argument available to explain differences in comparisons in the Arctic set, where 16 to 18 km is in the lower stratosphere (not the troposphere). During the summer (the time frame of the in situ Arctic measurements), there should be no clouds in the stratosphere for latitudes of 40 to 70 °N, so no possibility of them impacting the HDO fractionation. Over the years, there have been periodic volcanic eruptions that provided enhancements in stratospheric sulfate aerosols (liquid droplets of H2O and H2SO4 mixtures), which could provide a means of preferentially reducing atmospheric HDO through condensation, but volcanic eruptions that dramatically enhance stratospheric sulfates are not so frequent that they should pull the average significantly.

Looking at the excerpt of Figure 4 shown below, containing the Arctic results, the ~100 ‰ offset in the stratosphere cannot be explained away by clouds impacting the ACE results. Again, the most reasonable explanation would be that there is a systematic bias between the two instruments, some aspect in the analysis of one or both data sets that induces a systematic error in HDO fractionation, likely in the determination of HDO mixing ratio itself (rather than main isotopologue H2O). In my opinion, the strongest argument for the bias can be made by looking at pure stratospheric measurements in the Arctic set, but we also see similar offsets near 18 km for the other data sets in Figure 4, a persistent offset that strongly suggests issues in one or both data sets. Problems in the HDO line strengths wouldn't shock me, but there is no way to know for certain which data set is the biggest culprit in creating the bias.

Again, it would be appropriate to put stronger emphasis in the text on this systematic difference.

The authors have revised the relevant text to incorporate a bit more discussion about the Arctic HDO measurements, and to make it clear that there is likely a real bias between the two sampling methodologies, although it in not currently clear where the bias comes from.

Apologies if my previous feedback made the authors hesitant to highlight the discrepancy.

Minor comments
> Line 66: The instrument has been in operation from 2005 through the present day
The ACE-FTS has been in operation since 2004.
Corrected.

> Line 96: HOxotope
This instrument name is sometimes written as "Hoxotope" and other times as "Hoxotope."
The instrument name has been changed to 'Hoxotope' everywhere.

> Line 153: high-inclination (75◦) circular orbit

The inclination of the orbit is 74 degrees
Corrected.

> Line 157: The H2O molecule is ideally sampled from 5–150 km altitude
ACE-FTS H2O retrievals extend up to 95 km.
Corrected.

> Line 174: (RINSLAND et al., 1998)
All capital letters
Corrected.

> Index to Table 2 (and Tables 3 and 4): Counts represent seconds of sampling time for in situ instruments and number of occultations for ACE-FTS
There are no counts in these tables. There were counts in Table 1, so this is clearly a copy-and-paste issue.
Corrected.

---

## Author Response (AR2)

Reply to reviewer

In regards to the first correction, the authors did intend to refer to refer to Figure 1 on Line 136, and not Figure 2. To be a  bit more clear, the authors have amended the language here to refer to an 'average' instead of a 'climatology', as the latter implies more statistical detail than the authors provide.

The authors accept all other changes suggested by the reviewer, and have implemented them in the text.